# MULTI-SCALE LATENT POINT CONSISTENCY MODELS FOR 3D SHAPE GENERATION

## ABSTRACT

Consistency Models (CM) have significantly accelerated the sampling process in diffusion models, yielding impressive results in synthesizing high-resolution images. To explore and extend these advancements to point-cloud-based 3D shape generation, we propose a novel Multi-Scale Latent Points Consistency Model (MLPCM). Our MLPCM follows a latent diffusion framework and introduces hierarchical levels of latent representations, ranging from point-level to super-point levels, each corresponding to a different spatial resolution. We design a multi-scale latent integration module along with 3D spatial attention to effectively denoise the point-level latent representations conditioned on those from multiple super-point levels. Additionally, we propose a latent consistency model, learned through consistency distillation, that compresses the prior into a one-step generator. This significantly improves sampling efficiency while preserving the performance of the original teacher model. Extensive experiments on standard benchmarks ShapeNet and ShapeNet-Vol demonstrate that MLPCM achieves a 100x speedup in the generation process, while surpassing state-of-the-art diffusion models in terms of both shape quality and diversity.

## 1 INTRODUCTION

Generative modeling of 3D shapes plays a crucial role in various applications in 3D computer vision and graphics, empowering digital artists to create realistic, high-quality shapes. For these models to be practically effective, they must provide flexibility for interactive refinement, support the synthesis of diverse shape variations, and generate smooth meshes that seamlessly integrate into standard graphics pipelines. With the rapid advances in generative models for text, images, and videos, significant progress has also been made in 3D shape generation. Techniques based on variational autoencoders (VAEs) (Litany et al., 2018; Tan et al., 2018; Mittal et al., 2022), generative adversarial networks (GANs) (Wu et al., 2016; Shu et al., 2019; Hao et al., 2021), and normalizing flow models (Yang et al., 2019; Kim et al., 2020; Klokov et al., 2020) have been proposed to generate 3D shapes. Recently, diffusion-based models (Mo et al., 2023) and their latent diffusion variants (Zeng et al., 2022) have achieved state-of-the-art results in this domain.

Despite these advancements, challenges remain with applying diffusion models to 3D shapes represented as point clouds. First, the irregular spatial distribution and large number of points require the denoising model to capture both local and global geometric patterns accurately to produce high-quality 3D shapes. Diffusing in the latent space, where 3D shapes are summarized into more compact representations, is arguably more appealing than operating directly in data space, as it simplifies the learning process. However, prior work has demonstrated that relying on a single level (scale or spatial resolution) of latent representations is insufficient for achieving satisfactory performance. How to effectively leverage information from multiple levels remains a critical question. Second, the sampling process in diffusion models is notoriously slow, making them impractical for real-world applications.

In this paper, we introduce a novel model, Multi-scale Latent Points Consistency Models (MLPCM), which follows the latent diffusion framework. Our approach is a hierarchical VAE that incorporates multiple levels of latent representations, ranging from point-level to super-point levels, each corresponding to a different scale (spatial resolution). We establish a hierarchical dependency among these multi-scale latent representations in the VAE encoder. Diffusion is then performed on the

point-level latent representations, where noisy point-level latent inputs are modulated by latent representations from multiple super-point levels through a specially designed multi-scale latent integration module. This design allows our model to better capture both local and global geometric patterns of 3D shapes. To further improve performance, we incorporate a 3D spatial attention mechanism into the denoising function, where the bias term is determined by pairwise 3D distances, encouraging the model to focus on nearby regions within the 3D space. After training the model, we leverage consistency distillation to learn a latent consistency model, which significantly accelerates the sampling process.

In summary, our main contributions are as follows:

- We propose a novel Multi-scale Latent Points Consistency Models for 3D shape generation, which builds a diffusion model in the hierarchical latent space, leveraging representations ranging from point to super-point levels.

- We propose a multi-scale latent integration module along with the 3D spatial attention mechanism for effectively improve the denoising process in the latent point space.

- We explore the distillation of latent consistency models within the 3D latent point space, achieving fast (one-step), high-quality sampling.

- Extensive experiments show that our method outperforms exiting approaches on the widely used ShapeNet benchmark in terms of shape quality and diversity. Moreover, after consistency distillation, our model is 100x faster than the previous state-of-the-art in sampling.

## 2 RELATED WORKS

**Diffusion Models.** Diffusion models have seen significant success in image generation Ho et al. (2020); Song et al. (2020b; 2021); Song & Ermon (2019); Ramesh et al. (2022); Rombach et al. (2022); Nichol et al. (2021). These models are trained to denoise data that has been corrupted by noise, thereby estimating the score of the data distribution. During inference, they generate samples by running the reverse diffusion process, which gradually removes noise from the data points. Compared to VAEs Kingma (2013); Sohn et al. (2015) and GANs Goodfellow et al. (2020), diffusion models offer advantages in terms of training stability and more accurate likelihood estimation.

**Accelerating DMs.** Despite their success, diffusion models are limited by slow generation speeds. To address this, various approaches have been proposed. Training-free methods include ODE solvers Song et al. (2020b); Lu et al. (2022a;b), adaptive step-size solvers Jolicoeur-Martineau et al. (2021), and predictor-corrector methods Song et al. (2020b). Training-based strategies involve optimized discretizationWatson et al. (2021), truncated diffusionLyu et al. (2022); Zheng et al. (2022) , neural operatorsZheng et al. (2023), and distillation techniques Salimans & Ho (2022); Meng et al. (2023). Additionally, more recent generative models have been introduced to enable faster sampling Liu et al. (2022; 2023a). Song et al. (2023) have demonstrated that Consistency Models (CMs) hold great promise as a new type of generative model, offering faster sampling while maintaining high-quality generation. CMs utilize consistency mapping to directly map any point along an ODE trajectory back to its origin, allowing for rapid one-step generation. These models can be trained either by distilling pre-trained diffusion models or as independent generative models. Further details about CMs are provided in the following section.

**3D Point Cloud Generation.** This task is also referred to as shape generation. Achlioptas et al. (2018) were were the first to study this problem, developing a straightforward GAN with multiple MLPs in both the generator and discriminator. They also introduced several metrics to evaluate the quality of 3D GANs. Valsesia et al. (2018) improved the generator by incorporating graph convolution operations. Liu et al. (2018) employs a tree-structured graph convolution network (GCN) to capture the hierarchical information of parent nodes. Gal et al. (2021) uses a GAN with multiple roots to generate point sets that achieve unsupervised part disentanglement.

Beyond GAN-based methods, there are also approaches rooted in probability theory. Cai et al. (2020). proposed ShapeGF, which uses stochastic gradient ascent on an unnormalized probability density to move randomly sampled points toward high-density regions near the surface of a specific shape. DPM Luo & Hu (2021) conceptualized the transformation of points from a noise distribution to a point cloud as the inverse process of particle diffusion in a thermal system in contact with

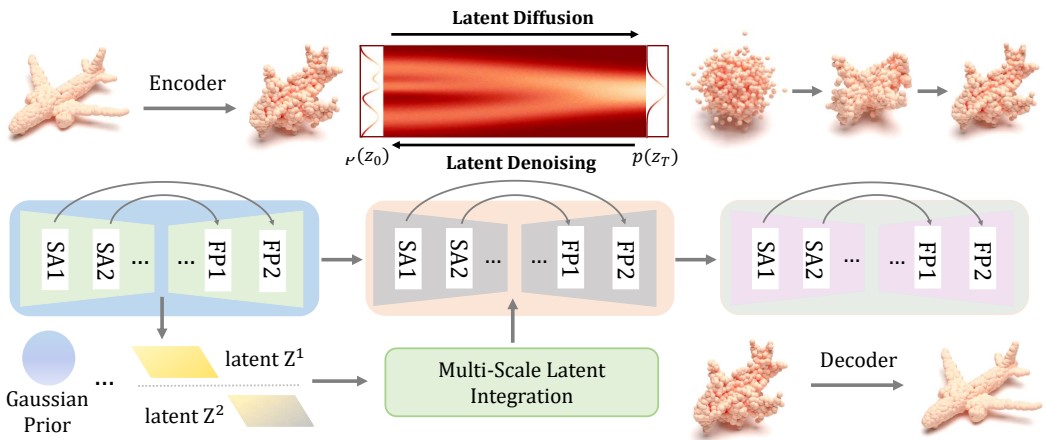

Figure 1: An overview of the proposed multi-scale latent diffusion model. Both the VAE net works and the latent point diffusion model are implemented based on Set Abstraction modules and Feature Propagation modules.

a heat bath. They modeled this inverse diffusion as a Markov chain conditioned on a particular shape. Yang et al. (2019) introduced PointFlow, which generates point sets by modeling them as a distribution of distributions within a probabilistic framework. Kimura et al. (2021) presented a flow-based model called ChartPointFlow, which constructs a map conditioned on a label, preserving the shape's topological structure. Li et al. (2022) developed a modified variational autoencoder for parts-aware editing and unsupervised point cloud generation.

Several methods have shown their effectiveness in the auto-encoding task using AE architectures, such as l-GAN Achlioptas et al. (2018), ShapeGF Cai et al. (2020), and DPM Luo & Hu (2021). However, despite advancements in point cloud generation, the implicit relationship between GANs and AEs, which offers significant prior information gain, remains largely unexplored.

## 3 MULTI-SCALE LATENT POINT CONSISTENCY MODELS

In this section, we will introduce our hierarchical VAE framework, the multi-scale latent point diffusion prior, and the latent consistency model.

### 3.1 HIERARCHICAL VARIATIONAL AUTOENCODER FRAMEWORK

We begin by formally introducing the latent diffusion framework, which is essentially a hierarchical VAE. We denote a point cloud as $X \in \mathbb{R}^{N \times 3}$, consisting of $N$ points with 3D coordinates. We then introduce a hierarchy of latent variables with different spatial scales/resolutions, denoting as $\mathcal{Z} = \{Z^0, Z^1, \ldots, Z^L\}$, where $L$ is the number of hierarchy levels. We use superscript to denote the index of scales. Here $Z^l \in \mathbb{R}^{N_l \times C_l}$ where $N_l$ and $C_l$ are the number of points and the number of channels at the $l$-th level latent space respectively. At the 0-th, we have $N_0 = N$ and $N_l > N_{l+1}$ for any $l = 0, \ldots, L - 1$. In other words, the $Z^0$ is latent point representation whereas $Z^l$ are latent super-point (*i.e.*, a subset of points) representations for any $l > 0$. The details of latent layers and points in each level are demonstrated in Sec. 4 .

The backbone of our hierarchical VAE consists of an encoder $q_\phi(\mathcal{Z}|X)$, a decoder $p_\psi(X|\mathcal{Z})$, and a prior $p(\mathcal{Z})$, where $\phi$ and $\psi$ are the learnable parameters. Specifically, our encoder is designed as follows,

$$q_\phi(\mathcal{Z}|X) := q_\phi(Z^L|X) \prod_{l=L-1}^{0} q_\phi(Z^l|Z^{l+1}, X), \qquad (1)$$

where $q_\phi(Z^L|X)$ and $q_\phi(Z^l|Z^{l+1}, X)$ are assumed to be factorized Gaussian distributions with learnable mean and fixed variance parameters. Our decoder and prior take the simple form

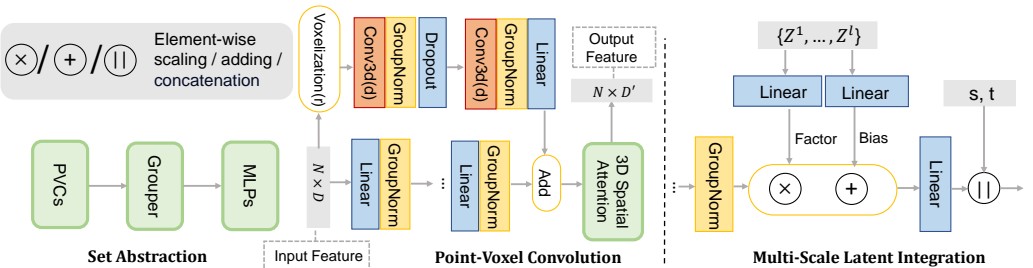

Figure 2: An illustration of the proposed network architecture. We denote the voxel grid size as $r$, and the hidden dimension as $D$ and $D'$.

$p_\psi(X|\mathcal{Z}) := p_\psi(X|Z^0)$ and $p(\mathcal{Z}) := \prod_{l=0}^{L} p(Z^l)$ respectively. Here $p(Z^l) := \mathcal{N}(\mathbf{0}, I)$. The decoder $p_\psi(X|Z^0)$ is parameterized as a factorized Laplace distribution with learnable means and unit scale parameters, corresponding to an L1 reconstruction error.

The architecture is illustrated in Fig. 1. As shown in the figure, both the encoder and the decoder consists of multiple Set Abstraction (SA) modules and Feature Propagation (FP) modules. Following Liu et al. (2019), each SA module contains point-voxel convolution layers and a Grouper block, where the Grouper block includes the sampling and grouping layers introduced by Qi et al. (2017b).

**First Stage Training.** We train the encoder and the decoder with a fixed prior by maximizing the modified evidence lower bound (ELBO),

$$\mathcal{L}_{\text{ELBO}}(\phi, \psi) = \mathbb{E}_{p_{\text{data}}(X), q_\phi(\mathcal{Z}|X)}[\log p_\psi(X|\mathcal{Z}) - \lambda D_{KL}(q_\phi(\mathcal{Z}|X)\|p(\mathcal{Z}))] \quad (2)$$

Here $p_{\text{data}}(X)$ represents the unknown data distribution of 3D point clouds. The hyperparameter $\lambda$ controls the trade-off between the reconstruction error and the Kullback-Leibler (KL) divergence.

## 3.2 MULTI-SCALE LATENT POINT DIFFUSION AS PRIOR

After learning the encoder and the decoder in the first stage, we fix them and train a denoising diffusion based prior in the latent point space. Note that we have a set of latent representations $\mathcal{Z} = \{Z^0, \ldots, Z^L\}$ ranging from point to super-point levels output by the hierarchical encoder. It is natural to select the latent variable with the coarsest scale (lowest spatial resolution), *i.e.*, $Z^L$, to build a diffusion prior since it would enable efficient diffusion in a low-dimensional latent space. However, as demonstrated in prior work (Zeng et al., 2022), point-level latent variables remain crucial for producing high-quality 3D shapes. We thus focus on the point-level latent variable $Z^0$ and design mechanisms to fuse multi-scale information. Specifically, we introduce a multi-scale latent point diffusion prior as below,

$$p(\mathcal{Z}) := p_\theta(Z^0|\mathcal{Z}^{\backslash 0}) \prod_{l=1}^{L} p_\theta(Z^l), \quad (3)$$

where $\mathcal{Z}^{\backslash 0} = \{Z^1, \ldots, Z^L\}$. Here $p_\theta(Z^l)$ is again a factorized standard Normal distribution, whereas $p_\theta(Z^0|\mathcal{Z}^{\backslash 0})$ is a diffusion model. Similar to other diffusion models, our model consists of the forward and the reverse processes.

**Forward Process.** Following diffusion models like DDPM (Ho et al., 2020), given initial latent point feature $Z^0 \sim q_\phi(Z^1|X)$ output by the encoder, we gradually add noise as follows,

$$q(Z_{1:T}^0|Z^0, \mathcal{Z}^{\backslash 0}) := \prod_{t=1}^{T} q(Z_t|Z_{t-1}^0), \qquad q(Z_t|Z_{t-1}^0) := \mathcal{N}(Z_t; \sqrt{1 - \beta_t} Z_{t-1}^0, \beta_t I), \quad (4)$$

where $Z_0^0 := Z^0$ and we use subscripts to denote diffusion steps. $T$ represents the number of diffusion steps, $q(Z_t|Z_{t-1}^0)$ is a Gaussian transition probability with variance schedule $\beta_1, \ldots, \beta_T$. We adopt a linear variance schedule in the diffusion process. The choice of $\beta_t$ ensures that the chain approximately converges to the stationary distribution, *i.e.*, standard Gaussian distribution $q(Z_T|Z^0, \mathcal{Z}^{\backslash 0}) \approx \mathcal{N}(Z_T; 0, I)$ after $T$ steps.

**Reverse Process.** In the reverse process, given the intial noise $Z_T^0 \sim \mathcal{N}(Z_T^0; \mathbf{0}, I)$, we learn to gradually denoise to recover the observed latent point feature $Z^0$ as follows,

$$p_\theta(Z_{0:T}^0 | \mathcal{Z}^{\backslash 0}) := p(Z_T^0) \prod_{t=1}^{T} p_\theta(Z_{t-1}^0 | Z_t^0, \mathcal{Z}^{\backslash 0}) \tag{5}$$

$$p_\theta(Z_{t-1}^0 | Z_t^0, \mathcal{Z}^{\backslash 0}) := \mathcal{N}(Z_{t-1}^0; \mu_\theta(Z_t^0, t, \mathcal{Z}^{\backslash 0}), \sigma_t^2 I), \tag{6}$$

where the mean function $\mu_\theta(\cdot, \cdot, \cdot)$ of the denoising distribution could be constructed by a neural network with learnable parameters $\theta$. In particular, following DDPM, we adopt the reparameterization $\mu_\theta(Z_t^0, t, \mathcal{Z}^{\backslash 0}) = (Z_t^0 - \beta_t \epsilon_\theta(Z_t^0, t, \mathcal{Z}^{\backslash 0}) / \sqrt{1 - \bar{\alpha}_t}) / \sqrt{\alpha_t}$, where $\alpha_t := 1 - \beta_t$ and $\bar{\alpha}_t := \prod_{s=1}^{t} \alpha_s$. The variance $\sigma_t$ is a hyperparameter and set to 1. Therefore, learning our multi-scale latent point diffusion prior is essentially learning the noise function $\epsilon_\theta(\cdot, \cdot, \cdot)$ parameterized by $\theta$.

**Second Stage Training.** To train the prior, we would ideally like to minimize the KL divergence between the so-called *aggregated posterior* $q(\mathcal{Z}) = \int p_{\text{data}}(X) q_\phi(\mathcal{Z}|X) \mathrm{d}X$ and the prior $p_\theta(\mathcal{Z})$. However, it is again intractable so that we need to resort to the negative ELBO, which can be equivalently written as the following denoising score matching type of objective,

$$\mathcal{L}_{\text{SM}}(\theta) = \mathbb{E}_{t \sim U([T]), p_{\text{data}}(X), q_\phi(\mathcal{Z}|X), \epsilon \sim \mathcal{N}(\mathbf{0}, I)} \| \epsilon - \epsilon_\theta(Z_t^0, t, \mathcal{Z}^{\backslash 0}) \|_2^2, \tag{7}$$

where $U([T])$ is the uniform distribution over positive integers $\{1, 2, \ldots, T\}$. Following Zeng et al. (2022), we utilize the mixed score parameterization, which linearly combine the input noisy sample and the noise predicted by the network, resulting in a residual correction that links the input and output of the noise function $\epsilon_\theta$. More details are provided in the appendix.

### 3.3 ARCHITECTURES OF MULTI-SCALE LATENT POINT DIFFUSION

We will introduce the specific architecture of our noise function $\epsilon_\theta$. Our model is based on Point-Voxel CNNs (PVCNNs) (Liu et al., 2019), similar to (Zhou et al., 2021; Zeng et al., 2022). Compared to 3D-Unet (Çiçek et al., 2016), PVCNN effectively handles large-scale point clouds by combining PointNet and voxel-based convolution (Qi et al., 2017a;b), capturing local relationships between points while preserving global information through the voxel convolutions. To integrate multi-scale latent point feature, we propose a multi-scale latent integration (MLI) module before each SA and FP module of PVCNNs. We also adopt a 3D spatial attention mechanism in each point-voxel convolution (PVC) module, allowing the model to selectively attend to informative areas, as shown in Fig. 2.

**Multi-Scale Latent Integration.** To better capture geometric patterns of 3D shapes, we propose the multi-scale latent integration (MLI) module to fuse latent variables across multiple scales/resolutions. As shown in Fig. 2, at the diffusion step $t$, the $s$-th MLI module takes the feature map $F_s$ as input, where $F_0 = Z_t^0$. Specifically, given the high-resolution feature map $F_s$ and the target feature $F$, we modulate the normalized target feature $F$ with the scaling parameters of both the two-dimensional scale and the high-scale features $F^i$, resulting in an intermediate representation as follows:

$$F_{s+1} = \left[ \text{MLP} \circ \text{ReLU}(\text{Linear}(\mathcal{Z}^{\backslash 0}) \odot \text{Norm}(F_s) + \text{Linear}(\mathcal{Z}^{\backslash 0})) \| \text{Pos}(t) \| \text{Pos}(k) \right], \tag{8}$$

where Norm is the layer normalization and Linear denotes a linear layer. MLP consists of two layers. $\odot$ denotes element-wise product and $\circ$ means function composition. $[\cdot \| \cdot]$ means concatenation of vectors. $\text{Pos}(\cdot)$ is the positional embedding function. To enable scale-time awareness in the model, we further modulate the feature with positional embedding of scale and time. The feature is fed to the following SA and FP modules to predict the noise.

**3D Spatial Attention Mechanism.** To help latent point representations better capture 3D spatial information , we design a 3D spatial attention mechanism. Our intuition is that closer points in 3D space would have a stronger correlation, thus higher attention values. We thus introduce a pairwise bias term that depends on the 3D distances between points. Specifically, in PVC, following Liu et al. (2019), we first use PointNet++Qi et al. (2017b) to extract the local features $F_p$ from the point cloud. We then convert the point cloud data into voxels, enabling the application of 3D convolution operations to obtain voxel features $F_v$. These are then fused to get the combined feature $F_a = \text{Add}(F_p, \text{MLP}(F_v))$. For $F_a$, we compute the pairwise spatial distance matrix $B \in \mathbb{R}^{n^i \times n^i}$

between different point pairs at the current scale, where $n^i$ is the number of points at the current scale $i$, and use it as a bias in computing the attention,

$$F_v = \text{softmax}\left(\frac{QK^T}{\sqrt{d}} + B\right) \cdot V \tag{9}$$

where $Q$, $K$, and $V$ are query, key, and value respectively in the standard attention module. We add this module to each PVC to make the model's attention conform to the 3D distances between points.

### 3.4 MULTI-SCALE LATENT POINT CONSISTENCY MODELS

Since the diffusion models are notoriously slow in sampling, we explore the consistency models (CMs) (Song et al., 2023) in the latent diffusion framework to accelerate the generation of 3D shapes. At the core of CMs, a consistency map is introduced to map any noisy data point on the trajectory of probability-flow ordinary differential equations (PF-ODEs) to the starting point, *i.e.*, clean data. There are two ways to train such a consistency map (Song et al., 2023), *i.e.*, consistency distillation and consistency training. The former requires a pretrained teacher model whereas the latter trains the consistency map from scratch. We explore both options in our context and found consistency distillation works well, whereas consistency is rather unstable.

Specifically, we first rewrite the reverse process in the setting of continuous time following (Song et al., 2020b; Lu et al., 2022a),

$$\frac{\mathrm{d}Z_t^0}{\mathrm{d}t} = \frac{\mathrm{d}\log\alpha_t}{\mathrm{d}t}Z_t^0 + \left(\frac{\mathrm{d}\log\alpha_t}{2\sigma_t\mathrm{d}t} - \frac{\mathrm{d}\log\alpha_t}{\mathrm{d}t}\sigma_t\right)\epsilon_\theta(Z_t^0, t, \mathcal{Z}^{\backslash 0}) \tag{10}$$

where $\epsilon_\theta(Z_t^0, t, \mathcal{Z}^{\backslash 0})$ is the noise prediction model and $\alpha_t, \sigma_t$ are determined by the noise schedule as aforementioned. Samples can be drawn by solving the PF-ODE from $T$ to $0$. To achieve consistency distillation, we introduce a consistency map $f_{\hat{\theta}} : (Z_t^0, t, \mathcal{Z}^{\backslash 0}) \mapsto Z_0^0$ to directly predict the solution of PF-ODE in Eq 10. For $t = 0$, we parameterize $f_{\hat{\theta}}$ using the noise prediction model $\hat{\epsilon}_\theta$ as follows:

$$f_{\hat{\theta}}(Z_t^0, t, \mathcal{Z}^{\backslash 0}) = c_{\text{skip}}(t)Z_t^0 + c_{\text{out}}(t)\left(\frac{Z_t^0 - \sigma_t\hat{\epsilon}_{\hat{\theta}}(Z_t^0, t, \mathcal{Z}^{\backslash 0})}{\alpha_t}\right), \tag{11}$$

where $c_{\text{skip}}(0) = 1$, $c_{\text{out}}(0) = 0$, and $\hat{\epsilon}_{\hat{\theta}}(z, c, t)$ is a noise prediction model whose initialized parameters are the same as those of the teacher diffusion model. Note that $f_{\hat{\theta}}$ can be parameterized in various ways, depending on the teacher model's parameterization of the diffusion model.

We assume that an effective ODE solver can be used to approximate the integral of the right side of Eq. 10 from any starting to ending time. Note that we only use the solver during training and not during sampling. We aims to predict the solution of the PF-ODE by minimizing the consistency distillation loss Song et al. (2023):

$$\mathcal{L}_{\text{CD}}(\hat{\theta}, \hat{\theta}^-; \Psi) = \mathbb{E}_{n, \mathcal{Z}^{\backslash 0}, Z_{t_{n+1}}^0}\left[d\left(f_{\hat{\theta}}\left(Z_{t_{n+1}}^0, \mathcal{Z}^{\backslash 0}, t_{n+1}\right), f_{\hat{\theta}^-}\left(\hat{Z}_{t_n}^0, \mathcal{Z}^{\backslash 0}, t_n\right)\right)\right], \tag{12}$$

where $d(\boldsymbol{x}, \boldsymbol{y}) = \|\boldsymbol{x} - \boldsymbol{y}\|_2^2$. $\hat{Z}_{t_n}^0 = Z_{t_{n+1}}^0 + (t_n - t_{n+1})\Psi(Z_{t_{n+1}}^0, t_{n+1}, t_n, \mathcal{Z}^{\backslash 0})$ is the solution of the PF-ODE obtained via calling the ODE solver $\Psi$ from time $t_{n+1}$ to $t_n$.

## 4 EXPERIMENT

In this section, we train the multi-scale latent diffusion model on the Shapenet dataset and obtain MLPCM using consistent distillation. We first introduce the dataset and evaluation metrics. Then we evaluate the performance of MLPCM on the Single-class 3D point cloud generation task. Next, we compare the results of MLPCM with a large number of baselines on many-class unconditional 3D shape generation, and compared the differences between MLPCM and other methods in sampling time and model parameters. We experimentally demonstrate that the proposed multi-scale latent diffusion model has achieved SOTA results under different experimental settings. MLPCM greatly improves the sampling speed while maintaining superior 3D point cloud generation results. Finally, we give detailed ablation experiments to test the effectiveness of multi-scale latents, 3D spatial attention bias, and the use of different extra dimension $D_h$ for the latent points.

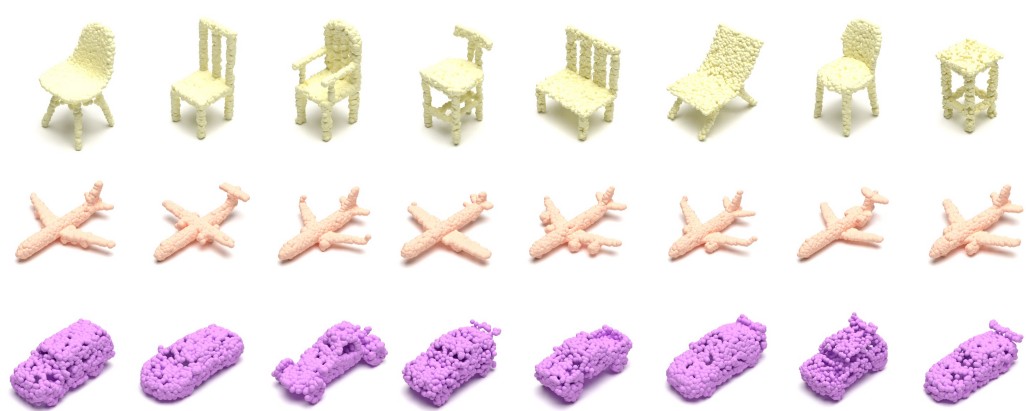

Figure 3: Qualitative visualizations of high-fidelity and diverse 3D point cloud generation.

## 4.1 DATASETS & METRICS

**Datasets** To compare MLPCM against existing methods, we use ShapeNet Chang et al. (2015), the most widely used dataset to benchmark 3D shape generative models. Following previous works Zeng et al. (2022); Yang et al. (2019); Zhou et al. (2021); Luo & Hu (2021), we train on three categories: airplane, chair, car. Also like previvous methods, we primarily rely on PointFlow's Yang et al. (2019) dataset splits and preprocssing. It normalizes the data globally across the whole dataset. However, some baselines require per-shape normalization Li et al. (2021); Zhang et al. (2021); Cai et al. (2020); Hui et al. (2020); hence, we also train on such data. In addition, following Zeng et al. (2022), we also perform the task of many-class unconditional 3D shape generation on the ShapeNet-Vol Peng et al. (2021; 2020) dataset. This data is also per-shape normalized.

**Evaluation** Model evaluation follows previous works Yang et al. (2019); Zhou et al. (2021). Various metrics to evaluate point cloud generative models exist, with different advantages and disadvantages, discussed in detail by Yang et al. (2019). Following recent works Yang et al. (2019); Zhou et al. (2021); Zeng et al. (2022), we use 1-NNA (with both Chamfer distance (CD) and earth mover distance (EMD)) as our main metric. It quantifies the distributional similarity between generated shapes and validation set and measures both quality and diversity Yang et al. (2019).

## 4.2 SINGLE-CLASS 3D POINT CLOUD GENERATION

**Implementation** Our implementation is based on the PyTorch framework Paszke et al. (2019). The input point cloud size is $2048 \times 3$. Both our VAE encoder and decoder are built upon PVCNN Liu et al. (2019), consisting of 4 layers of SA modules and 4 layers of FP modules. The voxel grid sizes of PVC at different scales are 32, 16, 8, and 8, respectively. We adopt the FPS (Farthest Point Sampling) algorithm for sampling in the Grouper block, with the sampled center points being 1024, 256, 64, and 16. KNN (K-Nearest Neighbors) is used to aggregate local neighborhood features, with 32 neighbors for each point in the Grouper. We apply a dropout rate of 0.1 to all dropout layers in the VAE. Following Zeng et al. (2022), we initialize our VAE model so that it acts as an identity mapping between the input, latent space, and reconstruction points at the start of training. We achieve this by reducing the variance of the encoder and correspondingly weighting the skip connections. Similarly, our Latent Points DDM Prior also consists of 4 layers of SA modules and FP modules. Notably, the time embeddings and multi-scale latents are concatenated with the point features at the input of each corresponding SA and FP layer, and we still use the hybrid denoising score network parameterization, similar to Zeng et al. (2022).

**Results** We use the Adam optimizer Kingma (2014) to train our model, where the VAE is trained for 8000 epochs and the diffusion prior for 24000 epochs, with a batch size of 128. We set $T = 1000$ as the time steps of the diffusion process. As for consistency distillation, we train MLPCM for 100K iterations with a batch size of 128, a learning rate of $2 \times 10^{-4}$, and an EMA rate $\mu = 0.99995$.

| Method | Airplane | | Chair | | Car | |
|---|---|---|---|---|---|---|
| | CD | EMD | CD | EMD | CD | EMD |
| r-GANAchlioptas et al. (2018) | 98.40 | 96.79 | 83.69 | 99.70 | 94.46 | 99.01 |
| l-GAN(CD)Achlioptas et al. (2018) | 87.30 | 93.95 | 68.58 | 83.84 | 66.49 | 88.78 |
| l-GAN(EMD)Achlioptas et al. (2018) | 89.49 | 76.91 | 71.90 | 64.65 | 71.16 | 66.19 |
| PointFlowYang et al. (2019) | 75.68 | 70.74 | 62.84 | 60.57 | 58.10 | 56.25 |
| SoftFlowKim et al. (2020) | 76.05 | 65.80 | 59.21 | 60.05 | 64.77 | 60.09 |
| SetVAEKim et al. (2021) | 76.54 | 67.65 | 58.84 | 60.57 | 59.94 | 59.94 |
| DPF-NetKlokov et al. (2020) | 75.18 | 65.55 | 62.00 | 58.53 | 62.35 | 54.48 |
| DPMLuo & Hu (2021) | 76.42 | 86.91 | 60.05 | 74.77 | 68.89 | 79.97 |
| PVDZhou et al. (2021) | 73.82 | 64.81 | 56.26 | 53.32 | 54.55 | 53.83 |
| LIONZeng et al. (2022) | 67.41 | 61.23 | 53.70 | 52.34 | 53.41 | 51.14 |
| MeshDiffusion Liu et al. (2023b) | 66.44 | 76.26 | 53.69 | 57.63 | 81.43 | 87.84 |
| DiT-3DMo et al. (2023) | 69.42 | 65.08 | 55.59 | 54.91 | 53.87 | 53.02 |
| Ours (Teacher Model) | **65.30** | **58.91** | **51.95** | **50.85** | **51.60** | **49.38** |
| Ours (Consitency Model) | 67.22 | 60.32 | 53.63 | 51.74 | 53.82 | 52.75 |

Table 1: Generation metrics (1-NNA↓) on airplane, chair, car categories from ShapeNet dataset from PointFlow Yang et al. (2019). Training and test data normalized globally into [-1, 1].

| Method | Airplane | | Chair | | Car | |
|---|---|---|---|---|---|---|
| | CD | EMD | CD | EMD | CD | EMD |
| Tree-GANLiu et al. (2018) | 97.53 | 99.88 | 88.37 | 96.37 | 89.77 | 94.89 |
| ShapeGFCai et al. (2020) | 81.23 | 80.86 | 58.01 | 61.25 | 61.79 | 57.24 |
| SP-GANLi et al. (2021) | 94.69 | 93.95 | 72.58 | 83.69 | 87.36 | 85.94 |
| PDGNHui et al. (2020) | 94.94 | 91.73 | 71.83 | 79.00 | 89.35 | 87.22 |
| GCAZhang et al. (2021) | 88.15 | 85.93 | 64.27 | 64.50 | 70.45 | 64.20 |
| LIONZeng et al. (2022) | 76.30 | 67.04 | 56.50 | 53.85 | 59.52 | 49.29 |
| Ours (Teacher Model) | **73.28** | **63.08** | **56.20** | **53.16** | 58.31 | 47.74 |
| Ours (Consitency Model) | 75.56 | 66.85 | 58.58 | 55.32 | 61.28 | 49.91 |

Table 2: Generation results (1-NNA↓) on ShapeNet dataset from PointFlow Yang et al. (2019). All data normalized individually into [-1, 1].

We validate our model's results under two settings: globally normalizing data across the entire dataset and normalizing each shape individually. Samples from MLPCM are shown in Fig. 3, and quantitative results are provided in Tab. 1 and Tab. 2. MLPCM outperforms all baselines and achieves state-of-the-art performance across all categories and dataset versions. Compared to key baselines like DPM, PVD, and LION, our samples are diverse and visually pleasing.

### 4.3 MANY-CLASS UNCONDITIONAL 3D SHAPE GENERATION

Following Zeng et al. (2022), we also jointly trained the multi-scale latent diffusion model and the MLPCM model across 13 different categories (airplane, chair, car, lamp, table, sofa, cabinet, bench, phone, watercraft, speaker, display, ship, rifle). Due to the highly complex and multi-modal data distribution, jointly training a model is challenging.

We validated the model's quantitative generation performance on the Shapenet-Vol dataset, where each shape is normalized, meaning the point coordinates are bounded within $[-1, 1]$. We report the quantitative generative performance of the model in Tab. 3, and trained various strong baseline methods under the same setting for comparison. We found that the proposed multi-scale latent diffusion model significantly outperforms all baselines, and MLPCM greatly improves sampling efficiency while maintaining performance.

| Method | CD(1- NNA↓) | EMD(1- NNA↓) |
|---|---|---|
| Tree-GANLiu et al. (2018) | 96.80 | 96.60 |
| PointFlowYang et al. (2019) | 63.25 | 66.05 |
| ShapeGFCai et al. (2020) | 55.65 | 59.00 |
| SetVAEKim et al. (2021) | 79.25 | 95.25 |
| PDGNHui et al. (2020) | 71.05 | 86.00 |
| DPF-NetKlokov et al. (2020) | 67.10 | 64.75 |
| DPMLuo & Hu (2021) | 62.30 | 86.50 |
| PVDZhou et al. (2021) | 58.65 | 57.85 |
| LIONZeng et al. (2022) | 51.85 | 48.95 |
| Ours (Teacher Model) | **50.17** | **47.84** |
| Ours (Consitency Model) | 53.85 | 52.45 |

Table 3: Generation results trained jointly on 13 classes of ShapeNet-vol.

| Method | steps | time(sec) | CD(1- NNA↓) | EMD(1- NNA↓) |
|---|---|---|---|---|
| LION(DDPM)Zeng et al. (2022) | 1000 | 27.09 | 53.41 | 51.14 |
| LION(DDIM)Zeng et al. (2022) | 1000 | 27.09 | 54.85 | 53.26 |
| LION(DDIM)Zeng et al. (2022) | 100 | 3.07 | 56.04 | 54.97 |
| LION(DDIM)Zeng et al. (2022) | 10 | 0.47 | 90.38 | 95.4 |
| Ours(Teacher Model) | 1000 | 25.16 | **51.60** | **49.38** |
| Ours(Consistency Model) | 1 | **0.18** | 79.46 | 82.00 |
| Ours(Consistency Model) | 4 | 0.31 | 53.82 | 52.75 |

Table 4: The sampling rates and generation quality of our model with various methods. Our approach achieves high-quality shape generation within 0.5 seconds while maintaining quality.

## 4.4 SAMPLING TIME

The proposed multi-scale latent diffusion model synthesizes shapes using 1000 steps of DDPM, while MLPCM synthesizes them using only 1-4 steps, generating high-quality shapes in under 0.5 seconds. This makes real-time interactive applications feasible. In Tab. 4, we compare the sampling time and quality of generating a point cloud sample (2048 points) from noise using different DDPM, DDIM Song et al. (2020a) sampling methods, and MLPCM. When using $\leq 10$ steps, DDIM's performance degrades significantly, whereas MLPCM can produce visually pleasing shapes in just a few steps.

## 4.5 ABLATION STUDY

In this section, we conduct ablation studies to demonstrate the effectiveness of the key 3D design components introduced in the multi-scale latent diffusion model, including the multi-scale latents, 3D Spatial Attention Mechanism, and the additional dimensions of the latent points.

**Ablation on the Key 3D Design Components** We perform an ablation experiment with the car category over the different components of the multi-scale latent diffusion model. For the setting without multi-scale latents, we consider two cases where only point-level and shape-level latent features are used. Please note that the model size remains nearly identical across different settings. The results in Tab. 5 show that the full setting, with multi-scale latents and the 3D spatial attention mechanism, performs the best across all metrics. This ablation study demonstrates the advantages of incorporating multi-scale latents and the 3D spatial attention mechanism.

**Ablation on Extra Dimensions for Latent Points**

We ablate the additional latent point dimension $D_h$ on the Car category in Tab. 6. We experimented with several different additional dimensions for the latent points, ranging from 0 to 5, where $D_h = 1$ provided the best overall performance. As the additional dimensions increase, we observe a general decline in the 1-NNA score. We use $D_h = 1$ for all other experiments.

| PL | SL | 3DSA | CD($\downarrow$) | EMD($\downarrow$) |
|----|----|------|------|-------|
|    | $\checkmark$ | $\checkmark$ | 74.29 | 72.81 |
| $\checkmark$ |    | $\checkmark$ | 53.40 | 51.96 |
| $\checkmark$ | $\checkmark$ |    | 53.02 | 50.75 |
| $\checkmark$ | $\checkmark$ | $\checkmark$ | **51.60** | **49.38** |

Table 5: Ablation studies on key 3D design components on the car category. PL and SL represent the point-level latents and shape-level latents, respectively, and 3DSA represents the 3D Spatial Attention Mechanism.

| Extra Latent Dim | CD($\downarrow$) | EMD($\downarrow$) |
|------------------|------|-------|
| 0 | 54.56 | 52.79 |
| 1 | **51.60** | **49.38** |
| 3 | 56.88 | 55.27 |
| 5 | 59.06 | 51.90 |

Table 6: Ablation on the number of extra dimensions $D_h$ of the latent points, on the car category.

## 5 CONCLUSION

In this paper, we propose the Multi-Scale Latent Point Consistency Model (MLPCM) to address the task of efficient point-cloud-based 3D shape generation. We first construct a multi-scale latent diffusion model, which includes a point encoder based on multi-scale voxel convolutions, a multi-scale denoising diffusion prior with 3D spatial attention, and a voxel-convolution-based decoder. The diffusion prior model, which integrates multi-scale information, effectively captures both local and global geometric features of 3D objects, while the 3D spatial attention mechanism helps the model better capture spatial information and feature correlations. Moreover, MLPCM leverages consistency distillation to compress the prior into a one-step generator. On the widely used ShapeNet and ShapeNet-Vol datasets, the proposed multi-scale latent diffusion model achieves state-of-the-art performance, and MLPCM achieves a 100x speedup during sampling while surpassing the state-of-the-art diffusion models in terms of shape quality and diversity. In the future, we are interested in extending MLPCM to other diverse 3D representations such as meshes.

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
