# OpenReview forum: "Multi-Scale Latent Points Consistency Models for 3D Shape Generation"
_ICLR.cc/2025/Conference — ICLR 2025 Conference Withdrawn Submission_

### Official Review · Reviewer_z8r2 · 2024-10-21

**Soundness:** 2
**Presentation:** 2
**Contribution:** 3
**Rating:** 5
**Confidence:** 4

**Summary:**

This paper proposes a novel Multi-scale Latent Points Consistency Models, which builds a diffusion model in the hierarchical latent space, leveraging representations ranging from point to super-point levels. This paper also proposees a multi-scale latent integration module along with the 3D spatial attention mechanism for effectively improving the denoising process in the latent point space.

**Strengths:**

The sampling acceleration presented in this paper is quite apparent, and the quantitative metrics indicate that the quality of the accelerated generation results is comparable to the baseline.

**Weaknesses:**

1. The claim that single-level representation is insufficient requires experimental evidence or appropriate citations.(# L45). The motivation and insight behind multi-level representation are not presented well throughout the paper.
2. Table 5 only provides a comparison for the car category. It would be helpful to have corresponding displays for the airplane and chair categories as well, or a joint result across all 13 classes, as ablation experiments on only a small set may be not convincing.
3. Visual comparison about baselines and ablation is absent. Visual comparison is very important when qualitatively understanding how multi-scale latent integration and 3DSA work.
4. Some recent works need to be discussed in the main paper, such as [1], [2], [3]

[1] SDFusion: Multimodal 3D Shape Completion, Reconstruction, and Generation

[2] 3DQD: Generalized Deep 3D Shape Prior via Part-Discretized Diffusion Process

[3] 3DILG: Irregular Latent Grids for 3D Generative Modeling

**Questions:**

1. #L287 is confusing. Do you mean to say that consistency training is unstable?
2. What do you mean by skip connection in #L370
3. What are the training and inference costs?
4. Why do DPM and MeshDiffusion show a much worse evaluation? Is there a possibility of an unfair comparison here?

**Details Of Ethics Concerns:**

No ethics review is needed.

---

### Official Review · Reviewer_LpEf · 2024-11-01

**Soundness:** 3
**Presentation:** 2
**Contribution:** 2
**Rating:** 6
**Confidence:** 4

**Summary:**

The paper proposes a latent consistency model for generating point clouds in a few steps and avoid the usual multi-step generation of the diffusion models. To better capture the 3D details and also overall shape of the objects, the diffusion process is applied on multiscale by grouping points together into structures called super points using a VAE architecture. In addition, a 3D spatial attention is used to make sure that points that are closer to each other attend more strongly. The quantitative results show improvement over other SOTA and some limited qualitiative results are also provided.

**Strengths:**

The paper is a contribution along the right direction. Diffusion models take time at inference and removing this burden helps their usage in more applications and helps users to interactively play with their results and select what they like.

There are certain aspects of the paper that I like. Although generation at multi-scale is not new, it is still the right choice. Also, I like the idea of 3D spatial attention in which closer points have higher correlations.

Its significant speed up in sampling without compromising quality too much is indeed impressive.

**Weaknesses:**

The paper is mostly adopted from previous works. In fact, the core contribution is the combination of consistency models with point cloud generation. This is mostly fine but makes me not get super excited about the paper but as I mentioned earlier, this is a step toward the right direction.

In fact, the two other contributions: 3D self attention and hierarchical point representation does not seem to help much. It is better to have only PL in Table 5 to have a better picture. It seems that the heavy lifting is done by PL and the rest just make the results slightly better, which is expected as PL has more info about the shape. But then, I am asking myself, the core contribution is really just consistency models applied on point clouds and others are marginal improvements.

It seems that the paper was written in rush. There are very few qualitative results and from the caption of Fig 3, it is not clear if the results are made in one step, through a teacher model or anything else. While, there is still space left in the paper, authors decided not to put more qualitative results under different settings and validate their work. Also, line 237 refers to the appendix but I was not able to find any appendix. Captions of the figures are also not expressive enough. For example caption of Fig 1 does not explain the components in the figure. Also, caption of fig 2 does not have explanation of the components. This is not a good practice in general.

There are also typos here and there e.g., line 98 were were, and line 71, exiting-> existing.

All in all, I am not very negative about the paper but I am not also very excited. I have to see other experts' opinions and make my final decisions.

**Questions:**

What happens if we don't use hierarchy of points at all? The results will get much worse?

Visually, how do the result look like when we 1, or 3 step generation is used?

What is the number of steps in consistency models reported in Tab 1 and 2? Authors should provide the numbers for different steps.

What are the limitations?

---

### Official Review · Reviewer_dq3n · 2024-11-01

**Soundness:** 2
**Presentation:** 2
**Contribution:** 2
**Rating:** 5
**Confidence:** 4

**Summary:**

In this paper, the authors propose a consistency model for generating 3D point cloud shapes, with a focus on achieving efficient, few-step generation. The architecture employs a multiscale hierarchical representation with a modified attention mechanism that prioritizes points in proximity, and the consistency technique is used to distill the model into a new one that operates in just a few steps (1-4). To the best of my knowledge, this is the first demonstration of distillation for a 3D point cloud generative model.

**Strengths:**

1. Following the general motivation of distillation into a few steps, the focus on few-step generation makes the model appealing for applications needing rapid feedback.
2. The multiscale hierarchical structure, along with the modified attention mechanism that prioritizes nearby points, helps capture 3D point cloud details at different scales, which should improve shape quality.
3. The experiments include both per-category and all-category training results, showing strong performance across different cases.

**Weaknesses:**

1. The paper doesn’t include specific modifications to the distillation process, making this contribution feel less novel.
2. Although few-step generation could theoretically support interactive editing, the authors don't provide a method for it. The model lacks specific design considerations for interactivity. Adding a conditioning approach, such as a voxel-based method similar to LION, could enable true interactivity and align better with the paper’s claims. It would also be interesting to see if the distillation process holds up under conditioning or guidance techniques.

**Minor:**
3. The related work section could be clearer and more focused on the paper’s main contributions.
4. The section on general diffusion models in related work doesn’t add much and could be streamlined.

**Typos:**
   - Line 286: "consistency training."
   - Line 305: "we aims"

**Questions:**

1. I didn't fully understand the explanation or motivations behind the MLI layer. Specifically, the choice to use latents to modulate features \( F_s \) lacks clear motivation. Additionally, the role of the two-dimensional scales isn’t clarified. More context on why this design was chosen would help.
2. The explanation of the VAE's latent variable structure is confusing, especially with respect to the latent numbering and the statement \( N_0 = N \). Equation (1) seems to indicate that \( X \) is encoded to \( Z^L \) rather than \( Z_0 \), which is contradictory. The figure also suggests that latents emerge from the encoder bottleneck, which does not match the text and adds to the confusion.
3. Figure 2 doesn’t show where MLI layers fit within the architecture, making it hard to follow their integration.
4. The bias term explanation is unclear. Since \( B \) is a distance matrix, it should be minimal for nearby points, but adding it as a dense map to the attention score would seem to emphasize distant points instead of close ones. Some clarification here would help.

---

### Official Review · Reviewer_3Mkj · 2024-11-02

**Soundness:** 3
**Presentation:** 2
**Contribution:** 2
**Rating:** 5
**Confidence:** 3

**Summary:**

The paper proposes Multi-scale Latent Points Consistency Models for 3D shape generation, which builds a diffusion model in the hierarchical latent space. Using a two-stage approach that first trains a hierarchical VAE to learn a hierarchical latent distribution from point-level to shape-level latent, a conditional diffusion model is then trained to sample the point-level latent with the information of the latent from a coarser level. Network architectures including a Multi-Scale Latent Integration and a  3D Spatial Attention Mechanism are used in the denoiser network to improve the sampling performance. A consistency model is further fit to sample efficiently from the diffusion model in a few steps. The paper shows impressive quantitative generation results compared to competitive baselines. Also, with the consistency model, the paper shows significant generation speed-up using the consistency model with minimal qualitative degradation. The ablation study also shows the importance of the hierarchical latent representation and the proposed network architecture.

**Strengths:**

1. Paper shows impressive quantitative results compared to many baselines in different generation setups. The paper evaluated their approach on single-category and all category generation and outperformed the baselines in the 1-NNA metric.
2. Paper shows significant speed-up using a consistency model with minimal performance degradation.
3. The proposed hierarchical latent point representation for point-cloud generation is effective and boosts the generation performance as shown in the ablation study.
4. The paper is clearly written and easy to follow.

**Weaknesses:**

1. The main claim is that the method enables point-cloud generation with a consistency model. However, it seems that the authors simply use the existing formulation of the consistency model without any modification. So why is this part a technical contribution of the paper? Why can't previous works like LION also incorporate a consistency model to speed up their diffusion sampling? While the sampling time comparison is impressive, the baseline LION only uses DDIM sampling. It would be great if the authors could clarify why the proposed method is better suited for consistency models than existing works.
2. The novelty of the proposed approach is limited. Hierarchical latent representation is not new and is explored in many other works to show improvement. The only modification of the proposed work from LION seems to be the usage of multiple latent hierarchies and the consistency model. However, for example, the recent CVPR 2025 paper XCube [1] also uses a hierarchical representation to improve generation results for large scenes. Consistency models can also be trained for these works to improve efficiency. The proposed network modifications are also not novel. The multi-scale latent integration looks similar to the AdaIn layer proposed in StyleGAN [2] and the spatial attention module is just an attention module with relative positional embedding.
3. Not enough qualitative examples. It seems that the supplementary is missing despite the promise in the paper of an appendix (c.f. ln 237). The paper only has one qualitative visual and they do not look impressive. Qualitative comparison with baselines should be provided.
4. It's not clear what the application that this method enables. Similar to LION, the method can only generate 2048 points for each of the shapes, and I'm not sure why this would be helpful for downstream applications, given the sparsity of the points. While LION did the same, they also had a downstream pipeline that could convert the point samples to meshes. I wonder if this method can do the same. Otherwise, it would be great if the authors could provide some applications enabled by this method.

[1] Xuanchi Ren, Jiahui Huang, Xiaohui Zeng, Ken Museth, Sanja Fidler, & Francis Williams. (2024). XCube: Large-Scale 3D Generative Modeling using Sparse Voxel Hierarchies.

[2] Tero Karras, Samuli Laine, & Timo Aila. (2019). A Style-Based Generator Architecture for Generative Adversarial Networks.

**Questions:**

1. Why the superscript to Z is dropped in distributions $q(Z_t|Z^0_{t-1})$ in Eq (4)?
2. Why $Z^0 \sim q_{\phi}(Z^1|X)$ in Ln 207?
3. In Eq (4) it seems that the forward process is independent of $\mathcal{Z}^{\backslash 0}$. So why write it there?

---

### Official Review · Reviewer_dB4Y · 2024-11-04

**Soundness:** 2
**Presentation:** 2
**Contribution:** 2
**Rating:** 5
**Confidence:** 3

**Summary:**

This paper proposes a new Multi-scale Latent consistency model to generate 3D point clouds and design a 3D spatial attention module to improve the performance. The authors also distill the trained consistency models into one-step generation to accelerate the sampling speed.

**Strengths:**

1. This paper explores to extend the consistency model to point cloud generation, which is useful for the 3D shape generation community.
2. The authors significantly accelerate the inference speed by adopting a distillation stage.
3. The proposed 3D attention module has proved effective.
4. The multi-scale representation is reasonable for point cloud generation.

**Weaknesses:**

1. The overall idea is a little incremental by using the consistency model to train 3D point cloud generation.
2. The effectiveness of 3D attention in point clouds has been explored in other point cloud tasks such as point transformers.
3. Although efficient and effective, the distillation stage is an additional engineering work to improve the sampling efficiency. Previous work like LION can also be accelerated by such distillation approaches and it is a little unfair to compare with LION with no additional distillation.

**Questions:**

See the weaknesses.

---

### Note · Authors · 2024-11-21

I have read and agree with the venue's withdrawal policy on behalf of myself and my co-authors.